# SARS-CoV-2 IgG Antibody Levels in Women with IBD Vaccinated during Pregnancy

**DOI:** 10.3390/vaccines10111833

**Published:** 2022-10-29

**Authors:** Irit Avni Biron, Yair Maayan, Tali Mishael, Eran Hadar, Michal Neeman, Romina Plitman Mayo, Hen Y. Sela, Simcha Yagel, Rosalind Goldenberg, Ami Ben Ya’acov, Sorina Grisaru Granovsky, Jacob E. Ollech, Hadar Edelman-Klapper, Keren Masha Rabinowitz, Maor H. Pauker, Henit Yanai, Sophy Goren, Dani Cohen, Iris Dotan, Ariella Bar-Gil Shitrit

**Affiliations:** 1Inflammatory Bowel Disease Center, Division of Gastroenterology, Rabin Medical Center, Petah-Tikva 4919001, Israel; 2Sackler Faculty of Medicine, Tel Aviv University, Tel Aviv 9436008, Israel; 3IBD MOM Unit, Shaare Zedek Medical Center, Digestive Diseases Institute, Jerusalem 9436008, Israel; 4Faculty of Medicine, The Hebrew University of Jerusalem, Jerusalem 9103102, Israel; 5Department of Military Medicine and “Tzameret”, Faculty of Medicine, Hebrew University of Jerusalem, Jerusalem, Israel, and Medical Corps, Israel Defense Forces, Rehovot 7661041, Israel; 6Obstetrics and Gynecology Department, Shaare Zedek Medical Center, Jerusalem 9103102, Israel; 7Helen Schneider Hospital for Women, Rabin Medical Center, Petah-Tikva 49100, Israel; 8Department of Immunology and Regenerative Biology, Weizmann Institute of Science, Rehovot 7610001, Israel; 9Obstetrics and Gynecology Department, Mount Scopus Hadassah Medical Center, Jerusalem 9103102, Israel; 10Felsenstein Medical Research Center, Sackler School of Medicine, Tel Aviv 69978, Israel; 11School of Public Health, Sackler Faculty of Medicine, Tel Aviv University, Tel Aviv 69978, Israel

**Keywords:** inflammatory bowel disease, pregnancy, COVID-19

## Abstract

Introduction: Regulatory agencies supported vaccination of pregnant women with SARS-CoV-2 mRNA vaccines, including patients with IBD. No data exist regarding these vaccines in IBD during pregnancy. Aim: To assess the serologic response to two doses of the mRNA SARS-CoV-2 BNT162b2 vaccine in pregnant women with IBD vaccinated during pregnancy, compared to that of pregnant women without IBD, and non-pregnant women with IBD. Methods: Anti-spike antibody levels were assessed in all women and in cord blood of consenting women. Results: From December 2020 to December 2021, 139 women were assessed: pregnant with IBD—36, pregnant without IBD—61, and not pregnant with IBD—42. Antibodies were assessed in cords of two and nine newborns of women with and without IBD, respectively. Mean gestational ages at administration of the second vaccine doses were 22.0 weeks in IBD and 23.2 weeks in non-IBD, respectively. Mean (SD) duration from the second vaccine dose to serology analysis in pregnant women with IBD, without IBD, and in non-pregnant women with IBD was 10.6 (4.9), 16.4 (6.3), and 4.3 (1.0) weeks, respectively. All women mounted a serologic response. In multivariable analysis, no correlation was found between the specific group and antibody levels. In both pregnancy groups, an inverse correlation between antibody levels and the interval from the second vaccine dose was demonstrated. Cord blood antibody levels exceeded maternal levels in women with and without IBD. Conclusion: All patients with IBD mounted a serologic response. The interval between vaccine administration to serology assessment was the most important factor determining antibody levels. A third vaccine dose should be considered in pregnant women with IBD vaccinated at early stages of pregnancy.

## 1. Introduction

Pregnant women were found to be at risk for severe COVID-19 complications and potentially for adverse pregnancy outcomes [1,2,3]. Several regulatory agencies and organizations, including the Centers for Disease Control and Prevention (CDC) and the American College of Obstetricians and Gynecologists (ACOG), have advised that the new mRNA COVID-19 vaccine should be offered to pregnant women [4,5,6,7,8]. As the prevalence of inflammatory bowel disease (IBD), including Crohn’s disease (CD) and ulcerative colitis (UC), is high in women of childbearing years [9], with IBD often being treated with immunosuppressive and biologic agents during pregnancy [10,11,12,13], pregnant patients with IBD constitute a unique group that requires special considerations. Phase 3 clinical trials conducted to evaluate the safety and efficacy of mRNA COVID-19 vaccines did not include pregnant women nor patients with IBD [14].

Emerging data on the administration of the COVID-19 vaccine during pregnancy has shown encouraging results in efficacy and safety [15,16,17,18]. In addition, receipt of a COVID-19 vaccine was shown to be immunogenic in pregnant women, and vaccine-elicited antibodies were transported to neonatal cord blood and breast milk [17,18].

IBD is characterized by chronic inflammation arising from an abnormal host immune response to dietary and microbial agents [19]. There is evidence that patients with IBD remain at significant risk of vaccine-preventable infections, suggesting vaccines confer suboptimal protection in this cohort [20,21,22,23]. Accumulating data in patients with IBD have shown that administration of the COVID vaccine to IBD patients is effective in prevention of infection [19,24]. Significant vaccine-related adverse effects and vaccine-related IBD exacerbation were not reported [25]. Serologic response to COVID-19 vaccines could be mounted in patients with IBD; however, its magnitude was significantly lower in patients treated with anti-tumor necrosis factor α (anti-TNF) agents regardless of administration timing and drug levels [26,27,28,29].

There is no data regarding the serological response to SARS-CoV-2 vaccine administration in patients with IBD during pregnancy. The objective of this study was to assess the serologic response to mRNA SARS-CoV-2 BNT162b2 vaccine in pregnant women with IBD, vaccinated during pregnancy, and to compare this serologic response to that of pregnant women without IBD and to that of non-pregnant women with IBD.

## 2. Materials and Methods

Study design and participants:

A multi-center study was conducted to compare the serologic response following the second dose of mRNA SARS-CoV-2 vaccine in three groups of women: pregnant women with IBD, pregnant women without IBD, and non-pregnant women with IBD.

Study Population:

The study population included women ≥18 years, who received two 30 µg BNT162b2 vaccine doses intramuscularly, 21–28 days apart, as per manufacturers’ recommendations. Women included in the study had no identified, reported previous COVID infection.

Three groups of women were compared:Pregnant women with IBD group: patients with an established diagnosis of IBD, followed up on and treated during pregnancy in dedicated IBD–Pregnancy joint clinics conducted at two medical centers in Israel (Shaare Zedek and Rabin medical center) between March and December 2021. As part of regular clinical care, consenting patients were evaluated in a prospective follow-up that included at least one visit per trimester, for clinical assessment and blood analysis. In consenting patients under biologic therapies, cord blood was also collected.Pregnant women without IBD—control group: consecutive parturient women, admitted for delivery at two tertiary medical centers in Israel (Shaare Zedek and Hadassah medical center) between January and August 2021. Eligible women were recruited on admittance to the delivery room or operating room in elective cesarean delivery cases to a prospective study aimed to assess maternal and neonatal SARS-CoV-2 immunoglobulin G (IgG) antibody levels.Non-pregnant women with IBD—control group: IBD female patients in reproductive age, included in a prospective study conducted at Rabin medical center, between December 2020 and May 2021, to assess response to the BNT162b2 mRNA COVID-19 vaccine in IBD patients and healthy controls. Non-pregnant women with IBD were selected according to age, IBD type, and anti-TNF therapy to compare with pregnant women with IBD.

### 2.1. Study Procedures and Data Collection

Pregnant women with IBD group: patients were evaluated at each pregnancy trimester for regular clinical assessment and follow-up. Laboratory tests were performed at each pregnancy trimester visit, including COVID-19 serology (SARS-CoV-2 IgG quantitative testing, anti-S and SARS-CoV-2 nucleocapsid (N) IgG). In consenting patients, an umbilical cord blood sample was obtained within 30 min of delivery. During regular follow-up visits in the joint IBD-pregnancy clinics, data regarding baseline demographics and IBD characteristics were collected, including age, BMI, parity, smoking status, comorbidities, IBD type and phenotype, IBD-related surgery, IBD-related medications during pregnancy, date and gestational age at the first and second vaccine doses and gestational age at birth. The time interval between the second vaccine dose and blood sample obtainment for COVID-19 serology was calculated.Pregnant women without IBD—control group: upon admittance to the delivery room or operating room in cases of elective cesarean delivery, maternal blood samples were obtained for COVID-19 serology (SARS-CoV-2 IgG quantitative testing, anti-S and SARS-CoV-2 nucleocapsid (N)-IgG). An umbilical cord blood sample was obtained within 30 min of delivery. After enrolment, demographic and clinical data were collected, including maternal age, BMI, parity, date and gestational age at the first and second vaccine doses and gestational age at birth. The time interval between the second vaccine dose and delivery was calculated.Non-pregnant women with IBD—control group: at enrollment, patients were assessed for baseline demographics and IBD characteristics. Blood samples for COVID-19 serology were obtained 21–35 days after the second vaccine dose (SARS-CoV-2 IgG quantitative testing, anti-S and SARS-CoV-2 nucleocapsid (N) IgG). The time interval between the second vaccine dose and blood sample obtainment for COVID-19 serology was calculated.

### 2.2. Laboratory Methods

Both maternal and umbilical cord blood samples were assessed using SARS-CoV-2 anti-S IgG II quantitative (anti-S levels) testing using the Abbott architect i2000sr platform following the manufacturer’s instructions. The SARS-CoV-2 IgG II quantitative assay is designed to detect IgG antibodies, including neutralizing antibodies to the receptor-binding domain of the S1 sub-unit of the spike protein of SARS-CoV-2 virus in human serum and plasma. Anti-S values ≥ 50 activity units (AU)/mL are considered positive.

SARS-CoV-2 nucleocapsid (N) IgG testing was performed semi-quantitatively using ELISA plates coated with N-protein following the manufacturer’s instructions (EUROIMMUN, Lubeck, Germany). Values ≥ 1.1 units are considered positive.

### 2.3. Outcomes

Our primary outcome was to assess seropositivity rate and magnitude of the serologic response (levels of binding immunoglobulin IgG antibodies to SARS-CoV-2 spike [S] antigen) following the second dose of BNT162b2 in pregnant women with IBD, compared with pregnant women without IBD, and with non-pregnant women with IBD. Several secondary outcomes were also assessed, including correlation between maternal characteristics and the serologic response (maternal age, gestational week at vaccination, time duration between 2nd vaccine and serological assessment), the correlation between the serologic response in pregnant women with and without IBD, and according to specific IBD characteristics (IBD type, anti TNF therapy), the correlation between the serologic response in IBD pregnant and non-pregnant women with IBD, and the assessed maternal and cord blood serologic response in women with and without IBD.

### 2.4. Statistical Analysis

Anti-S antibody concentrations were expressed as geometric mean concentrations (GMCs) with 95% confidence intervals (CIs). After data collection, the correlation between antibody titers, maternal and IBD characteristics and the time interval from the 2nd vaccine dose to serological assessment were analyzed. Continuous variables were presented as means and SDs or as medians and IQRs. Categorical variables were presented as percentages. Student’s t-test or one-way ANOVA were used to compare continuous variables, whereas non-parametric distributed data were analyzed using the Mann–Whitney or Kruskal–Wallis test. To estimate correlations with maternal antibody levels, we used a univariable linear model with a logarithmic transformation for the antibody level because it was not normally distributed, as indicated by the Kolmogorov–Smirnov test. Correlation coefficients (r) within 95% confidence intervals for the correlation between the maternal antibody level and duration from receipt of the second vaccine dose were presented. Variables that were found to be statistically significant (2-sided *p* < 0.05) were entered into a multivariable regression model to estimate adjusted associations with antibody levels. In case of significant difference between baseline parameters, variables were controlled using regression models.

## 3. Results

### 3.1. Patients

From December 2020 to December 2021, 139 women were recruited: IBD pregnant women group—36, non-IBD pregnant women control group—61, and non-pregnant women with IBD group—42. Antibody titers to COVID vaccine were measured in all 139 women and in cord blood of two and nine newborns of women with and without IBD, respectively. The participants’ baseline characteristics are presented in Table 1.

Mean age was comparable between groups (32.1 ± 5.5, 29.4 ± 5.5, and 31.5 ± 6.5, *p*-0.073). Gestational age and gestational weight at delivery were also comparable (38.8 ± 1.9 weeks, 39 ± 2.6 weeks, 0.570 and 3135 ± 527.8 g, 3245 ± 523.7 g, *p*-0.390 respectively). Rates of cesarean delivery were higher in the women with IBD as compared with women without IBD (IBD-21 (61.8%) vaginal delivery and 13 (38.2%) cesareans, non-IBD-55 (93.2%) vaginal delivery and 4 (6.8%) cesareans, *p* < 0.001).

Pregnant women with IBD included 22 (61.1%) women with Crohn’s disease (CD) and 14 (38.9%) women with ulcerative colitis (UC). Anti-TNF therapy was administered during pregnancy to 11 (30.6%) women. IBD characteristics including medical therapy in pregnant women with IBD, and non-pregnant women with IBD are presented in Table 2.

### 3.2. Vaccine Administration and Response

The mean gestational ages at administration of the first and second vaccine doses were 19.7 weeks and 22.0 weeks in the IBD group and 20.1 weeks and 23.2 weeks in the non-IBD group, respectively.

The mean (SD) duration from the second vaccine dose to serology analysis in pregnant women with IBD, without IBD, and in non-pregnant women with IBD was 10.6 (4.9), 16.4 (6.3), and 4.3 (1.0) weeks, respectively.

All maternal anti-S levels were positive. The GMC IgG levels (95% CI) in pregnant women with IBD, without IBD, and in non-pregnant women with IBD were 2442 (1489–4004), 977 (716–1332), 7416 (4966–11,073), respectively (Table 3).

Anti-N, reflecting previous infection with COVID-19, was negative in all patients but one pregnant woman without IBD; this patient was included in the analysis.

Among pregnant women with IBD, no significant difference was demonstrated in the GMC (95% CI) anti-S levels in accordance with anti-TNF treatment (2408 (718–8071) in anti-TNF treated women vs. 2457 (1419–4253) in women not treated with anti TNF). No significant difference was demonstrated in GMC (95% CI) anti-S levels in accordance with IBD type (1900 (1061–3403) in CD and 3620 (1403–9337) in UC).

Univariable analysis demonstrated correlation of anti-S levels with specific cohort group, gestational age at 2nd vaccine dose administration, and the duration from the second vaccine dose to serology analysis. For each week that passed following the second vaccine administration, ln of anti-S levels changed on average by −0.176 (CI: −0.260 to −0.092, *p* < 0.001) in women with IBD and by −0.095 (CI: −0.138 to −0.051, *p* < 0.001) in women without IBD.

Multivariable analysis showed no correlation between the specific cohort groups (Pregnant with and without IBD, and IBD with and without pregnancy), and anti-S levels (Table 4). Furthermore, in both pregnancy groups (with and without IBD), multivariable analysis revealed an inverse correlation between anti-S levels and the time interval from the second vaccine dose (Figure 1 and Figure 2). For each week that passed following the second vaccine dose, ln of antibody level changed on average by −0.124 (CI: −0.162 to −0.086, *p* < 0.001 regardless of the group (with or without IBD) (Table 4).

Serological response was assessed in the cord blood of 11 newborns, 2 born to women with IBD and 9 without IBD. Cord blood of all newborns tested positive for anti-S levels.

Neonatal anti-S levels GMC = 2003.0 (95% CI: 877–4574.8) significantly exceeded maternal anti-S levels GMC = 609.4 (95% CI: 286.5–1296) (*p* = 0.028, by *t*-test) in both newborns of women with and without IBD (*n* = 11).

## 4. Discussion

In this study of patients vaccinated with two doses of the SARS-CoV-2 mRNA BNT162b2 vaccine, serological response was assessed in a group of women with IBD vaccinated during pregnancy and compared with two groups of patients: pregnant women without IBD and non-pregnant women with IBD.

In women with IBD, receipt of two doses of vaccine during the second trimester of pregnancy was associated with a serologic response in all women, placental transfer of antibodies to cord blood was demonstrated in all neonates, and cord blood levels exceeded maternal levels. In a multivariable regression analysis comparing the three groups of patients, latency between vaccine administration and serologic assessment was significantly associated with the level of serologic response, whereas carrying a diagnosis of IBD or being pregnant were not.

The importance of latency from vaccination to serological assessment as a major contributor to antibody levels has been well demonstrated in previous publications, in the general population [30,31,32,33], in immune-suppressed populations including patients with IBD [34,35], as well as in cohorts of pregnant women vaccinated during pregnancy [36,37,38]. A systematic review of 18 studies involving 15,980 participants demonstrated that the levels of protective antibodies significantly and consistently declined 6–8 months after the second dose of vaccine, regardless of patient-related factors and peak antibody levels [31]. Similar reduction in antibody levels with time was also well described in a systematic review involving 9447 IBD patients [34]. In a prospective cohort of 129 pregnant women vaccinated with two doses of vaccine during the second trimester, for each week that passed following receipt of the second vaccine dose, maternal antibody levels changed by −10.9% [36]. A significant reduction in antibody levels from 2–6 weeks after vaccination to delivery was also demonstrated by Ayeto et al. in 26 women vaccinated during the first and second trimesters of pregnancy [38].

The current study included cord blood analysis of nine newborns to women without IBD and two newborns with IBD. Maternal to neonatal transfer was demonstrated in all cases, as described in several previous publication, showing high levels of placental transfer that approach 100% [37,38,39,40].

In all 11 neonatal-maternal dyads assessed in the current study, neonatal antibody levels exceeded maternal levels. Interestingly, a comparable neonatal/maternal antibody ratio of 2.6 was reported by Kugelman et al. in an analysis of 114 pregnant women vaccinated during the second trimester and their neonates [36]. A median placental transfer ratio in the range of 0.77 to 2.6 was reported in additional studies, implying that the mechanisms and factors that affect the rate and amount of placental transfer of the SARS-CoV-2 vaccine are not fully understood [37,41].

In an evaluation of the neonatal to maternal transfer according to the timing of vaccination during pregnancy, Ayeto et al. demonstrated lower maternal and neonatal absolute titers, but higher neonatal to maternal ratios when vaccination occurred in the first and second trimesters. These findings suggest high placental efficiency in the setting of early pregnancy vaccination but waning maternal titers by delivery [38]. Whether higher antibody transfer ratios observed for early pregnancy vaccination are due to differences in antibody Fc-quality, leading to improved trafficking of antibodies to the fetal circulation, or are the result of increased time for antibody transit to occur, or a combination of both, is yet to be determined [38].

In the two newborns to women with IBD reported in our study, neonatal and maternal levels resembled those of the non-IBD population. Further larger studies are needed to confirm this observation in the IBD population.

Anti-TNF therapy has been associated with a reduced serological response to the SARS-CoV-2 vaccine in IBD patients in several publications [26,29].

Furthermore, in an evaluation performed six months after vaccination, patients with IBD treated with anti-TNF demonstrated significantly impaired serological responses, specifically, more seronegativity, decreased specific circulating B cells and cross-reactivity compared to patients untreated with anti-TNF [42].

In the current cohort of women with IBD, 30% of whom were treated with an anti-TNF, no significant difference was demonstrated in the serological response between CD and UC patients or between women treated and not treated with anti-TNF. This was true for both pregnant and non-pregnant women with IBD, probably due to the small number of women evaluated.

Here, we describe maternal and neonatal response to SARS-CoV-2 vaccine in pregnant women with IBD for the first time. A comparison of the serologic response to that of pregnant women without IBD and to non-pregnant women with IBD was also performed. Data reported in this study can contribute to the discussion with pregnant IBD patients, suggesting that no obvious, noticeable differences were demonstrated, compared to the general population or IBD population, in the serologic response to the vaccine during pregnancy. Several limitations need to be acknowledged. Data was extracted from three different separate cohorts, and serological assessment was performed at different timepoints. These differences could influence results and limit generalizability. A multivariable regression model was used to mitigate this limitation. Cord blood serological assessment was performed in a small number of neonates, only two with IBD precluding a firm conclusion. SARS-CoV-2 protection rate among groups and adverse events related to vaccination were not reported.

## 5. Conclusions

In conclusion, all woman with IBD, vaccinated with two doses of SARS-CoV-2 vaccine during pregnancy, mounted a serological response, and transplacental transfer to the cord blood was demonstrated. The time period between vaccine administration to serology assessment appeared to be the most important factor determining the serological response, supporting consideration of a third dose of vaccine in pregnant women with IBD, vaccinated at early stages of pregnancy.

## Figures and Tables

**Figure 1 vaccines-10-01833-f001:**
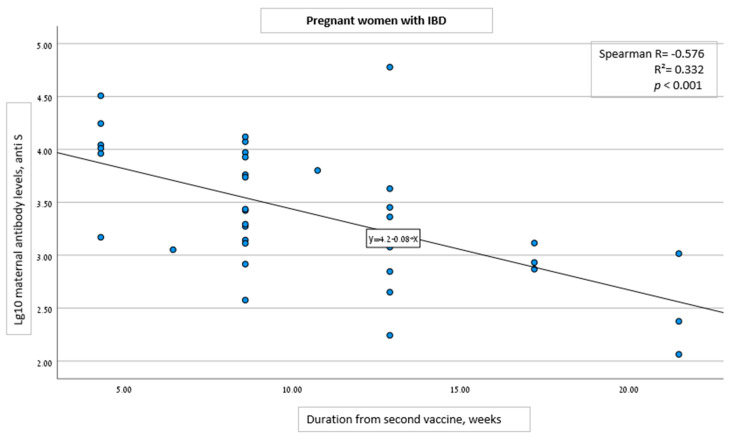
Correlation between the time interval from the second COVID-19 vaccine dose and SARS-CoV-2 IgG antibody level in pregnant women with IBD.

**Figure 2 vaccines-10-01833-f002:**
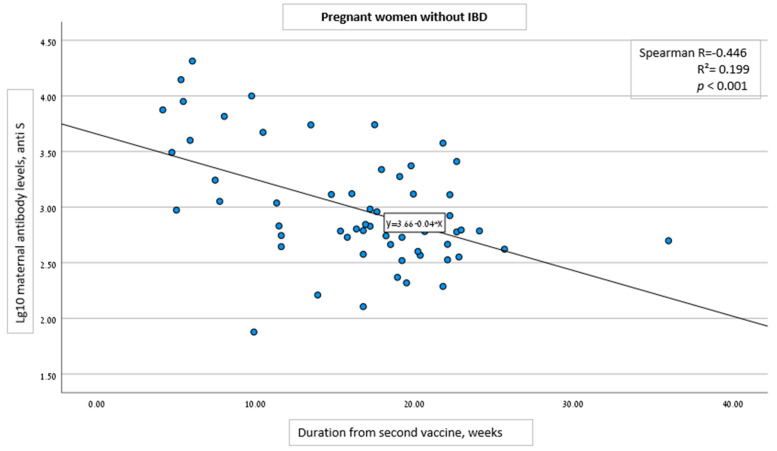
Correlation between the time interval from the second COVID-19 vaccine dose and SARS-CoV-2 IgG antibody level in pregnant women without IBD.

**Table 1 vaccines-10-01833-t001:** Demographics and clinical characteristics of women vaccinated with BNT162b2 mRNA COVID-19 vaccine.

	Pregnant with IBD	Pregnant without IBD	Non-Pregnant, IBD	*p* Value
Characteristic	N = 36	N = 61	N = 42	
Mean age, years (SD)	32.1 (5.5)	29.4 (5.5)	31.5 (6.5)	0.073
Mean weight, kg (SD)	70.8 (15.1)	64.8 (13.1)	61.3 (11.6)	**0.012**
Mean BMI, kg/m^2^ (SD)	26.9 (4.9)	24.2 (4.7)	23.6 (4.6)	**0.004**
Parity (SD)	2.3 (2.1)	2.2 (1.9)	0.9 (1.5)	**<0.001**
Smoking:				
Never	32 (88.9%)	59 (98.3%)	37 (88.1%)	0.085
Past/present	4 (11.1%)	1 (1.7%)	5 (11.9%)	
IBD type:				
Crohn’s disease	22 (61.1%)	-	29 (69.0%)	0.484
Ulcerative colitis	14 (38.9%)	-	13 (31.0%)	
Mean CRP (SD)	1.1 (1.8)	-	0.8 (1.5)	**0.006**
Anti-TNF:				
No	25 (69.4%)	-	29 (69.0%)	1.000
Yes	11 (30.6%)	-	13 (31.0%)	
Gestational age at delivery, mean (SD)	38.8 (1.9)	39.1 (2.6)		0.570
Mode of delivery:				
Vaginal	21 (61.8%)	55 (91.7%)	-	**<0.001**
Cesarean	13 (38.2%)	5 (8.3%)	-	
Birthweight, mean (SD)	3135 (527.8)	3235 (527.2)	-	0.390
Mean gestational age at 1st vaccine dose administration (weeks)	19.7 (8.8)	20.1 (6.7)		0.786
Mean gestational age at 2nd vaccine dose administration (weeks)	22.0 (8.2)	23.2 (6.8)		0.464
Time duration from 2nd vaccine to serology assessment, weeks, (SD)	10.6 (4.9)	16.4 (6.3)	4.3 (0.99)	**<0.001**

**Table 2 vaccines-10-01833-t002:** IBD characteristics and medical therapy.

	Pregnant with IBD (N = 36)	Non-Pregnant with IBD (N = 42)
Mean age, years (SD)	32.1 (5.5)	31.5 (6.5)
Mean BMI, kg/m^2^ (SD)	26.9 (4.9)	23.6 (4.6)
Smoking status:		
Present/past	4 (11.1%)	5 (11.9%)
No	32 (88.9%)	37 (88.1%)
Comorbidities	7 (19.4%)	12 (28.6%)
IBD-related surgery	8 (22.9%)	-
IBD phenotype:		
CD	22 (61.1%)	29 (69%)
UC	14 (38.9%)	13 (31%)
Medical therapy:		
Infliximab	5 (13.9%)	5 (11.9%)
Adalimumab	6 (16.7%)	9 (21.4%)
Vedolizumab	4 (11.1%)	8 (19.0%)
Ustekinumab	2 (5.6%)	4 (9.5%)
5-ASA	15 (41.7%)	12 (28.6%)
Corticosteroids	2 (5.6%)	7 (16.7%)
6mp/azathioprine	8 (22.2%)	5 (11.9%)
JAK inhibitor	0 (0%)	4 (9.5%)
No medical therapy	4 (11.1%)	3 (7.1%)

**Table 3 vaccines-10-01833-t003:** SARS-CoV-2 IgG antibody levels, GMC anti-S (95% CI).

	Pregnant with IBD	Pregnant without IBD	Non-Pregnant with IBD	*p* Value
SARS-CoV-2 IgG antibody levels, GMC anti-S (95% CI)	N = 36	N = 61	N = 42	
Total	2442 (1489–4004)	977 (716–1332)	7416 (4966–11,073)	<0.001
Anti-TNF				
No	2457 (1419–4253) *n* = 25	-	8920 (5812–13,690) *n* = 29	
Yes	2408 (718–8071) *n* = 11	-	4911 (1915–12,595) *n* = 13	
IBD type				
Crohn’s disease	1900 (1061–3403) *n* = 22	-	7601 (4679–12,346) *n* = 29	
Ulcerative colitis	3620 (1403–9337) *n* = 14	-	7018 (3081–15,985) *n* = 13	

**Table 4 vaccines-10-01833-t004:** Multivariate linear regression analysis of SARS-CoV-2 immunoglobulin G antibody level.

Variable	Change in Antibody Level% (95% CI)	*p* Value
Pregnant women with vs. without IBD		
Per 1-week increase from 2nd vaccine dose to serology analysis	−0.124 (−0.162 to −0.086)	<0.001
Group		
Pregnant IBD	0.090	0.365
Pregnant without IBD	reference	
Pregnancy week at 2nd vaccine	−0.121	0.262
IBD pregnant vs. IBD non-pregnant		
Per 1-week increase from 2nd vaccine dose to serology analysis	−0.172 (−0.233 to −0.110)	<0.001
Group		
IBD pregnant	−0.021	0.876
IBD non-pregnant	reference	
**Variable**	**Change in antibody level%** **(95% CI)**	***p* value**
**Pregnant women with vs. without IBD**		
**Per 1-week increase from 2nd vaccine dose to serology analysis**	−0.124 (−0.162 to −0.086)	**<0.001**
**Group**		
Pregnant IBD	0.090	0.365
Pregnant without IBD	reference	
**Pregnancy week at 2nd vaccine**	−0.121	0.262
**IBD pregnant vs. IBD non-pregnant**		
**Per 1-week increase from 2nd vaccine dose to serology analysis**	−0.172 (−0.233 to −0.110)	**<0.001**
**Group**		
IBD pregnant	−0.021	0.876
IBD non-pregnant	reference	

## Data Availability

The data underlying this article are available in the article.

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
