# Peer review of "SARS-CoV-2 IgG Antibody Levels in Women with IBD Vaccinated during Pregnancy"

_vaccines, 2022, doi:10.3390/vaccines10111833_

Round 1
Reviewer 1 Report
Authors studied SARS-CoV-2 IgG antibody in women with IBD vaccinated during pregnancy.
Overall, the study is not novel. They only compared antibody level in women with pregnant women with IBD group, pregnant women without IBD, and non-pregnant women with IBD. They found that antibody level is similar among groups. I recommend authors to include the data on SARS-CoV-2 protection rate among groups.
Minor
Authors had better discuss the reason of higher antibody in cord blood than in mom
Reviewer 2 Report
The key point of this paper is to compare the level of the Spike-specific antibody among pregnant with IBD, pregnant without IBD and not pregnant with IBD. But I can not see any conclusion on this.
Major points:
1. The Non-Pregnant women's data in line 211 is different from the data listed in the Table 3. Which one is correct?
2. Compared the S-specific antibody, in Pregnant without IBD is lower than Pregnant with IBD? Could the author explain this?
Minor points:
1. Some of the citation formats are wrong, like, [1 2, 3] and [1718]. The author needs to revise them.
Reviewer 3 Report
In introduction part the authors should write more about the influence of therapy on the serologic response in this type of patients. In discussion part I belive it is important that autohors to compare the influence of therapy with other study in this kind of patients. The authors should highlight the limitations of the study. The conclusions are inssuficient.
Reviewer 4 Report
The manuscript describes the anti-SARS-CoV-2 humoral responses after the vaccination with BNT162b2 in pregnant women with IBD. Although it may provide information important for this group of patients, it has a number of flaws described below.
As suggested by the authors, antibody levels are not normally distributed, however I was not able to find a description of statistical assessment of normality nor a mode of description of not-normally distributed variables. Please also verify and describe the statistical tests used in relation to normality of variables.
As the time duration from 2nd vaccine to serology assessment is a factor highly influencing a level of antibodies, the large difference between the groups strongly influences the results.
In the methods the authors mentioned SARS-CoV-2 nucleocapsid (N) IgG testing - It would be important as a proof of no previous infection, however I was not able to find the results.
Table 2 - please provide the detailed title and include the results of group comparisons
Table 3 - please provide the detailed title, units and provide the results of group comparisons
Fig. 1 and 2 - the values on the Y scale does not match the results presented in the text and tables
Table 3 - please provide the detailed title and units
“Neonatal anti S levels measured approximately 2.6 and 3.25 times higher than 255 maternal titers in newborns to mothers without IBD and with IBD respectively” - please provide the statistical significance of the observation.
“Authors should discuss the results and how they can be interpreted from the perspective of previous studies and of the working hypotheses. The findings and their implications should be discussed in the broadest context possible. Future research directions may also be highlighted” - interesting part of the conclusions
Problem with reference numbering throughout the manuscript (missing brackets, commas, spaces)
Round 2
Reviewer 1 Report
Authors responded well to my two questions. I recommend this manuscript to be accepted in Vaccines
Author Response
we thank the reviewer for the time and effort invested in helping us improve our manuscript.
Reviewer 2 Report
The authors addressed all of my questions.
Minor comment:
1. Make it consistent with "IgG", but not "igG" in the manuscript.
Author Response
we thank the reviewer for the time and effort invested in helping us improve our manuscript. we have reviewed the manuscript and corrected inconsistencies regarding IgG
Reviewer 3 Report
I dont have any other comments.
Author Response
We thank the reviewer for the time and effort invested in helping us improve our manuscript.
Reviewer 4 Report
Most of my remarks have been correctly addressed by the authors.
However, small changes are still required:
It's not clear how the antibody data are presented - "The median(95%CI) GMC IgG levels" - median or geometric mean? The authors should take care of the values and their notations throughout the text . It is also important to include p value, wherever any difference or similarity between the values is suggested.
“Anti-N, reflecting previous infection with COVID-19, was negative in all patients but one pregnant woman without IBD” - was she included or excluded from the group? A comment on that is missing.
Author Response
We would like to thank the reviewer for the time and effort spent in helping us improve out manuscript.
It's not clear how the antibody data are presented - "The median(95%CI) GMC IgG levels" - median or geometric mean?
Thank you for this important comment. Antibodies were presented as mean geometric concentration, the text was reviesed and corrected accordingly- in lines 213,221,223,264-5 (highlighted).
It is also important to include p value, wherever any difference or similarity between the values is suggested.
Thank you for this important comment. P values were added, lines 193,195,198- (highlighted).
Anti-N, reflecting previous infection with COVID-19, was negative in all patients but one pregnant woman without IBD” - was she included or excluded from the group? A comment on that is missing.
we thank the reviewer for this comment. A comment was added to the text in line 217. (highlighted).